# Cultural Itineraries Generated by Smart Data on the Web

**Cosmo Capodiferro** [1], **Massimo De Maria** [2], **Mauro Mazzei** [1,*], **Matteo Spreafico** [1], **Oleg V. Bik** [2], **Armando L. Palma** [1] and **Anna V. Solovyeva** [2]

1 National Research Council, Istituto di Analisi dei Sistemi ed Informatica, LabGeoInf, Via dei Taurini, 19,
I-00185 Rome, Italy; cosmo.capodiferro@iasi.cnr.it (C.C.); palma@arpal.it (A.L.P.)

2 Academy of Engineering, Peoples Friendship University of Russia (RUDN University),
6 Miklukho-Maklaya Street, 117198 Moscow, Russia; de_mm@pfur.ru (M.D.M.); bik-ov@rudn.ru (O.V.B.);
solovyeva-anv@rudn.ru (A.V.S.)

* Correspondence: mauro.mazzei@iasi.cnr.it

**Abstract:** The development of storage standards for databases of different natures and origins makes it possible to aggregate and interact with different data sources in order to obtain and show complex and thematic information to the end user. This article aims to analyze some possibilities opened up by new applications and hypothesize their possible developments. With this work, using the currently available Web technologies, we would like to verify the potential for the use of Linked Open Data in the world of WebGIS and illustrate an application that allows the user to interact with Linked Open Data through their representation on a map. Italy has an artistic and cultural heritage unique in the world and the Italian Ministry of Cultural Heritage and Activities and Tourism has created and made freely available a dataset in Linked Open Data format that represents it. With the aim of enhancing and making this heritage more usable, the National Research Council (CNR) has created an application that presents this heritage via WebGIS on a map. Following criteria definable by the user, such as the duration, the subject of interest and the style of the trip, tourist itineraries are created through the places that host this heritage. New possibilities open up where the tools made available by the Web can be used together, according to pre-established sequences, to create completely new applications. This can be compared to the use of words, all known in themselves, which, according to pre-established sequences, allow us to create ever new texts.

**Keywords:** linked open data; SPARQL; GeoSPARQL; geocoding; OpenStreetMap; web-GIS

## 1. Introduction

Our contribution in this paper is primarily aimed at interlinking data from different sources so that they can be linked and interconnected. This approach allows for a more extensive and complete view of information. Our proposal of Linked Open Data (LOD) is an initiative that aims to make data available in an open, interoperable, and linked way using web standards and semantic technologies. The added value with these methods is related to data reuse, providing a way to make data accessible and reusable by a wide range of users, promoting information sharing and reusability. The availability of linked data can improve the quality of online research by providing more detailed and connected information. The goal of this work is to make geospatial data open and accessible, allowing different parties to use and share information related to geographic location. Open applications of GIS data can encourage the development of geospatial information, e.g., applications for navigation in various modes, spatial analysis, urban planning, environmental monitoring, etc. The advantage of having GIS data available is to support location-based decisions by providing accurate and up-to-date geospatial data on which to base decisions on policy, planning, and land management. Open access to geospatial data can promote innovation and research in areas such as tourism, culture, urban planning, geography, natural resource management, and more, fostering the involvement of the scientific, cultural, citizen, developer and

organization communities in the creation and use of geospatial data. These motivations encouraged us to develop and experiment with a platform that would consider the aspects that are very significant to us and make a strong contribution to smart data in the geospatial domain. In the following, we will give an overview of the most popular open standards systems. Our approach also focuses on Semantic Web vision techniques, which seek to add semantic meaning to data to improve machine understanding. The semantic structure of linked data can be exploited for the development of more intelligent and contextually aware applications.

### 1.1. Semantic Web

The Semantic Web provides a common framework that allows data to be shared and reused across application, enterprise, and community boundaries. It is a collaborative effort led by W3C with participation from a large number of researchers and industrial partners.

In computer science, Linked Data defines a group of principles to be followed to publish and connect structured data on the Web so that they are understandable by computers and whose meaning is explicitly defined [1].

The principles are as follows:

- Use URI as name for objects;
- Use HTTP URIs so that people can access these names;
- Provide useful information using standards (RDF, SPARQL) when accessing a URI;
- Include links to other URIs so you can discover other objects.

It is therefore clear that Linked Data depends on two fundamental technologies for the Web: Uniform Resource Identifier (URI) [2] and HyperText Transfer Protocol (HTTP), which use the Resource Description Framework (RDF) format [3] to create typed links between arbitrary objects.

In RDF, data is organized into triplets composed of subject, predicate and object. The subject and object can either be a URI representing a resource or a URI and a text string. The predicate indicates how the subject and object are related and is also identified by a URI [4].

Using HTTP URI to identify resources, the HTTP protocol to retrieve them and the RDF data model to represent them, we can therefore consider Linked Data as an additional layer built on top of the general Web architecture [5], which maintains the same characteristics:

- Generic and can contain any type of data;
- Anyone can publish data;
- You are not constrained and a predefined set of relationships;
- Entities are connected with RDF, creating a global graph that includes multiple data sources and allows you to discover new ones.

The basic Resource Description Framework (RDF) format tool proposed by W3C for the encoding, exchange and reuse of structured metadata allows semantic interoperability between applications that share information on the Web.

As we have already mentioned, the RDF format allows us to represent Linked Data by expressing the type of connection between the data [6], but it does not allow us to assign an ontological meaning to the terms that are associated with the triples. We must therefore add a further level composed of ontologies and vocabularies in order to bring out the meaning of the relationships, thus obtaining what is called the Semantic Web [7].

Although there is no clear distinction between "ontology" and "vocabulary", the term "ontology" tends to be used for more complex and articulated knowledge organization systems with a more rigorous formal structure [8].

The standards promoted by the W3C as data models for the semantic web are Web Ontology Language (OWL) [9] and Simple Knowledge Organization (SKOS) [10]; the former is more formal, rigorous, and complex than the latter, but they can be used jointly depending on the scenarios [11].

The current orientation for the definition of the semantic web involves a bottom-up approach [12] according to the collaborative, flexible and free spirit of the web as an alternative to the centralized control of information.

### 1.2. Linked Open Data

Linked Data is not always freely usable or accessible on the Web, or in other circumstances the data is published freely, such as Open Data, but does not meet the characteristics of being Linked Data.

To better classify data sources, Tim Berners-Lee introduced a five-star rating system [1] as reported in Table 1.

**Table 1.** World Wide Web Consortium (W3C)-developed standards and guidelines.

| Ranking | Description |
|---|---|
| * | Available on the web (whatever format) but with an open license, to be Open Data; |
| ** | Available as machine-readable structured data (e.g., excel instead of an image scan of a table); |
| *** | The same as (2) plus a non-proprietary format (e.g., CSV instead of excel); |
| **** | All the above, plus uses open standards from W3C (RDF and SPARQL) to identify things, so that people can point at your stuff; |
| ****** | All the above, plus links your data to other people's data to provide context. |

When the published data obtains a five-star rating, we can properly speak of Linked Open Data [13], and it is possible to fully benefit from the Network Effect [14].

### 1.3. SPARQL

To be able to query Linked Data, the RDF Data Access Working Group, which is part of the W3C Semantic Web Activity, developed the SPARQL language [15]. It is a query language for data represented through the Resource Description Framework (RDF), whose queries are composed of three parts [16]:

– The pattern matching part supports several features, including optional parts, pattern merging, nesting, value filtering, and the ability to select the data source to use.
– The part of solution modifiers, where it is possible to modify the data selected from the previous part by applying the classic projection, uniqueness, ordering and limit operators.
– The output part of a SPARQL query can be of different types: yes/no, selection of variable values, creation of a new RDF from the data and description of the resources.

### 1.4. WebGIS

The term WebGIS or World Wide Web Geographic Information Systems indicates the process that allows cartographic data to be published on the Web so that they can be consulted and queried [17].

Compared to traditional cartography, they have countless advantages but also disadvantages [18]:

- They contain updated data. Maps on the Web contain data extracted in real time from databases and do not need to be printed and distributed.
- The hardware and software infrastructure for publishing maps on the Web is very cheap, while the production of geographic data is expensive, and for this reason, it is not always openly available.
- They can aggregate data from different sources. Being published on the Web, they can exploit its architecture to access and aggregate data from various sources on the Web.
- They are collaborative. All Web users can participate in integrating and correcting the data, but as in any collaborative project, close attention must be paid to the quality of the data and its origin.

- They support hyperlinks. Like any other resource on the Web, they may contain links to other resources on the Web.

### 1.5. Functions

By relating different data on the basis of their common geographical reference, it is possible to create new information starting from existing data, thanks to the wide possibilities of interaction that this technology offers.

Among the various possibilities, we list some of the most common ones [19]:

- Terrain analysis: various aspects concerning the terrain can be shown on the map to facilitate analysis such as inclination, orientation, and visibility between various points, but also the change in appearance due to human interventions or natural disasters.
- Distance analysis: techniques for calculating and optimizing distance-related cost functions, such as optimal route calculation.
- Data analysis: by crossing different thematic data on the map, for example through the representation of isolines, additional information can be obtained.
- Geostatistical analyses: algorithms for analyzing the spatial correlation of georeferenced variables.
- Data mining: search for hidden patterns in large databases, for example for environmental monitoring.
- Geocoding: obtaining the spatial coordinates of an address or place.

### 1.6. Geocoding

We do not always have the spatial coordinates of the elements we want to represent on the map available, so in this case, we must resort to a process that allows us to obtain the geographical coordinates starting from the description or address of a place [20].

This process presents several critical issues:

- Retrieval of address data;
- Verification of their correctness;
- Localization and verification of their updating over time.

Different standards are used around the world to encode addresses, and this does not facilitate analysis and recognition; input errors made by the user or the possibility that partial addresses are provided make geolocalization complex when there are multiple similar addresses in the various databases.

To improve this process, thanks to the development of Deep Learning and Computer Vision, it is possible to use technologies based on object recognition to identify the roofs of buildings [21] and, recently, researchers have also created a web crawler to extract spatial data from the internet by analyzing, for example, websites with real estate adverts [22].

The opposite process, which is called reverse geocoding, allows you to obtain a list of places close to the geographical coordinates provided. There are growing concerns regarding privacy due to the increasing ease of access to geocoding and "reverse geocoding", so it is advisable to be careful and possibly mask part of the information published to appropriately protect privacy.

### 1.7. Critical Issues

For a system to understand the intended meaning of information present in other systems, the information exchanged must be accompanied by an interoperable description. When using Linked Data, this description is found in linked anthologies that define the meaning of some properties and classes. The property of exchanging ontologies and using them is called semantic interoperability [23].

The difficulty in achieving semantic interoperability derives from the degree of heterogeneity of the semantics used by systems, which can have various origins:

- Structural conflicts, when the same (or overlapping) classes or properties are represented differently in two ontologies due to the different level of generalization/specialization used [24].
- Data conflicts, when the same concept is represented in a different way due to typing errors or the use of different identification systems.
- Logical conflicts, when there are differences in the way in which ontologies choose to organize concepts using, for example, different subclasses of the same class or separate classes [25].

A further critical issue arises from the open nature of Linked Open Data, where "anyone can say anything about anything" which raises a problem of trust relating to the sources of information [26].

## 2. Related Works

This section examines case studies that use Linked Open Data and represents them on thematic maps via WebGIS.

### 2.1. Cammini D'Italia (Atlas of Paths)

The Cammini d'Italia (Atlas of Paths) project [27] is an initiative of the National Research Council (CNR), with the aim of designing and implementing ontological modules organized in a knowledge graph that models the domain of paths. An application has been designed that allows users to implement the path domain through guided data entry and their subsequent automatic verification.

Subsequently, it is thought that it will be possible to design and implement a modular software platform for the generation, management and use of Linked Data and the semantic technologies associated with them [28].

#### 2.1.1. Architecture

The architecture used for this project [29] is illustrated in Figure 1.

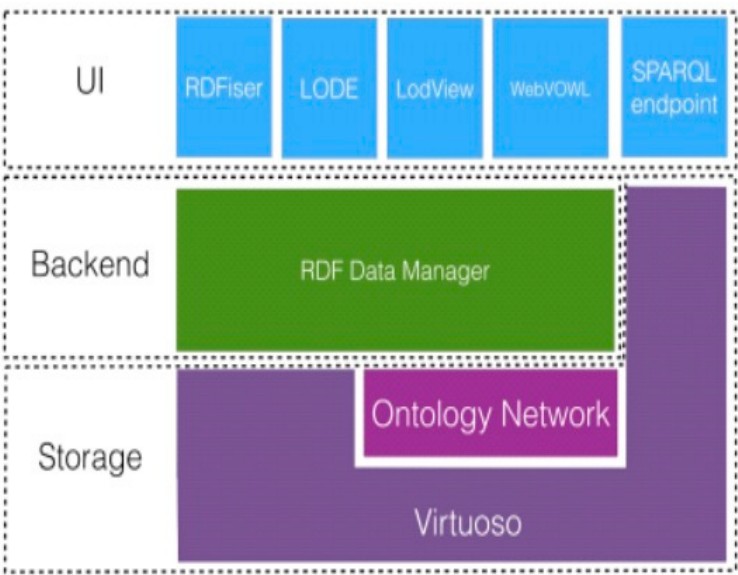

**Figure 1.** Architecture of Project "Cammini d'Italia".

The data is saved in the Storage layer in a triple store, which is responsible for persisting it in a long-lasting manner as well as providing Create, Read, Update and Delete (CRUD) functions and allowing you to query the data using SPARQL. It also contains the Ontology Network, i.e., the set of ontologies used by the project.

At the Backend level, we find the RDF Data Manager, which allows the insertion and querying of data as Open Linked Data in RDF format by the various modules of the UI level, namely:

- RDFiser gives a user the possibility to create Linked Open Data by filling in web forms;
- The Live OWL Documentation Environment (LODE) automatically generates documentation in HTML format for ontologies [30];
- LodView is a viewer of RDF data in HTML format [31];
- WebVOWL is a web application for the interactive and exploratory visualization of ontologies that uses the Visual Notation for OWL Ontologies (VOWL) [32] as notation.

2.1.2. Ontologies

To achieve the objective of creating a conceptual model of the route network, including their regional connections and crossings, it was necessary to define two new ontologies: Routes [33] and Atlas of Paths [34].

The Routes module defines general concepts and relationships to describe routes, stages and travel plans of any type; therefore, it acts as a support module for Atlas of Paths, which is instead the domain-specific ontological module.

For example, the Pathway concept defined in Atlas of Paths is a specialization related to the Routes path domain, which defines generic paths.

In addition to defining new ontologies, the project also uses other existing Linked Data, such as:

- The ArCo project [35], consisting of ontologies that describe the different types of cultural heritage and the cataloging records associated with them and managed by the Central Institute for Cataloging and Documentation (ICCD) [36];
- The Food in Open Data (FOOD) project [37], consisting of the reference ontologies for the semantic representation of the specifications relating to the quality brands of agri-food products made available by the Ministry of Agricultural, Food and Forestry Policies (MiPAAFT).

*2.2. EcoDigit*

The Digital Ecosystem project for the use and enhancement of Lazio's cultural assets and activities (EcoDigit) is one of the initiatives of the Center of Excellence of the Technological District for Cultural Assets and Activities (DTC), made up of the five state universities of Lazio, La Sapienza, Tor Vergata, Roma Tre, Cassino and Tuscia, in network with CNR, ENEA and INFN to aggregate and integrate skills in the sector of technologies for cultural heritage and activities [38].

The LOD paradigm was used to connect data from different cultural institutions. Collaboration between cultural organizations led to the collaborative development of ontologies describing cultural heritage internationally, such that semantic interoperability requirements could be met within their systems. Furthermore, the use of common ontologies has facilitated data exchange such as, for example, the European Data Model (EDM) [39].

The EcoDigit project led to the development of:

- A reference architecture for the integration of modular services and for their publication and reuse;
- A software component developed for the orchestration of services, the integration and aggregation of interfaces and data;
- A prototype version of services oriented towards the use and valorization of cultural heritage.

2.2.1. Architecture

The EcoDigit reference modular architecture [40] is represented in Figure 2. and is composed of five main areas:

- The Data Layer, for storing, indexing and searching data;
- The Acquisition Layer, to acquire data from various sources and appropriately convert them, where necessary, into Linked Data;
- The Middleware, which allows Client Applications to carry out semantic queries of contents and establishes guidelines for interoperability with external services;
- The Client Application, which represents the applications that consume EcoDigit contents via the Middleware;
- The Presentation Layer, which offers the possibility to navigate and manage the data present in EcoDigit, as well as the system administration, support and analysis functions.

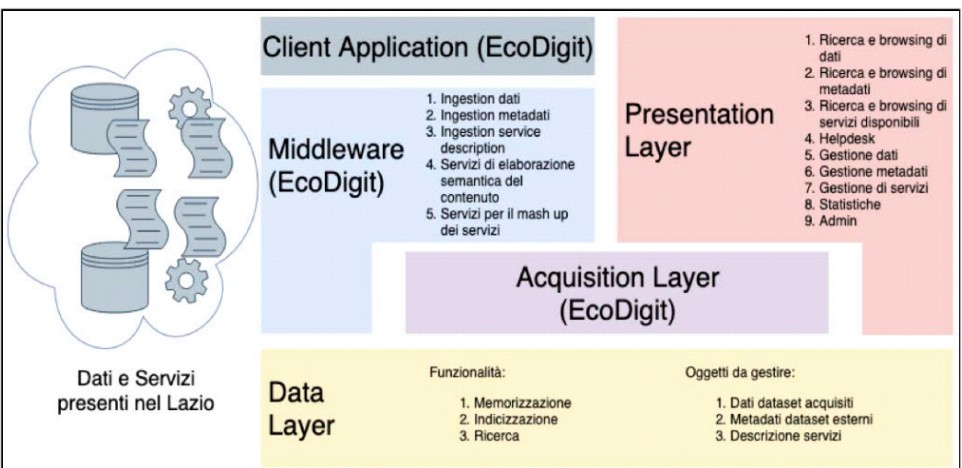

**Figure 2.** Architecture of Project "EcoDigit".

### 2.2.2. Ontologies

The data managed by EcoDigit follows the ontologies recommended by national bodies such as AgId [41] and international W3C or recognized as de facto standards such as OntoPiA, DOAP, The Organization Ontology [42], SPAR Ontologies [43], ArCo, and others.

Due to the peculiarity of the data processed, it was also necessary to define new ontologies that extend the existing ontologies where necessary [44]:

- Ontology of organizations

    Namespace: https://w3id.org/ecodigit/ontology/organization/ (accessed on 10 December 2023).

    Objective: the ontology aims to define a shared vocabulary of terms for the description of the organizations participating in the Center of Excellence—DTC Lazio.

- Ontology of experiences and skills

    Namespace: https://w3id.org/ecodigit/ontology/eas/ (accessed on 10 December 2023). Objective: the ontology aims to define a shared vocabulary of terms for describing a person's experiences and skills.

- Ontology of evaluations

    Namespace: https://w3id.org/ecodigit/ontology/grade/ (accessed on 10 December 2023).

    Objective: the ontology aims to define a vocabulary of terms for the description of anything that has an associated rating expressed on a certain scale.

- Project ontology

    Namespace: https://w3id.org/italia/onto/Project (accessed on 10 December 2023).

    Objective: the ontology aims to define a computational model for data relating to public projects.

2.2.3. Prototypes

EcoDigit has created and made prototypes available both to adhere to their data entry model in the project and to create clients that allow the use of the contents. As regards data input, a prototype is available for the selective collection of Linked Open Data [45], while the client prototype, using semantic technologies, allows for the search for resources and their connection to the contents of thematic maps (GIS) and 3D reconstructions to allow their use in virtual environments [46].

## 3. Case Study: Euplea Framework

The application implemented is a Progressive Web App created in TypeScript using the Next.js framework. On first access, a Service Worker is installed on the user's device, and thanks to the presence of a Web App Manifest, the user is asked whether he wishes to install the application on his device, mobile or desktop.

The REST APIs it uses are implemented within the Next.js framework using TypeScript and run within Node.js, a JavaScript runtime-built JavaScript V8 from Google Chrome.

The decision to use Next.js was dictated by the desire to be able to implement both the client and server components within the same project, using the same programming language. Moreover, during the development of the project, this made it possible not only to simply share software modules, but also to be able to move application logics from the client to the server in a completely transparent and immediate manner, making it possible to identify and define a valid compromise for the construction of REST APIs potentially reusable in other projects and the realization of a Progressive Web App.

Instead of JavaScript, we decided to use TypeScript because the latter language was built as a superset of JavaScript with the addition of an optional static typing system that allows static analysis of the code and the detection and avoidance of many of the most common errors by the compiler, providing a smoother and faster programming experience.

The architecture is depicted in Figure 3, and shows how a single artefact dealing with the creation of the itinerary has been realized, which receives user requests and consumes the services offered by Cultural Heritage and OpenStreetMap, to generate an itinerary to be shown to the user by exploiting a caching system implemented via Azure Cosmos DB.

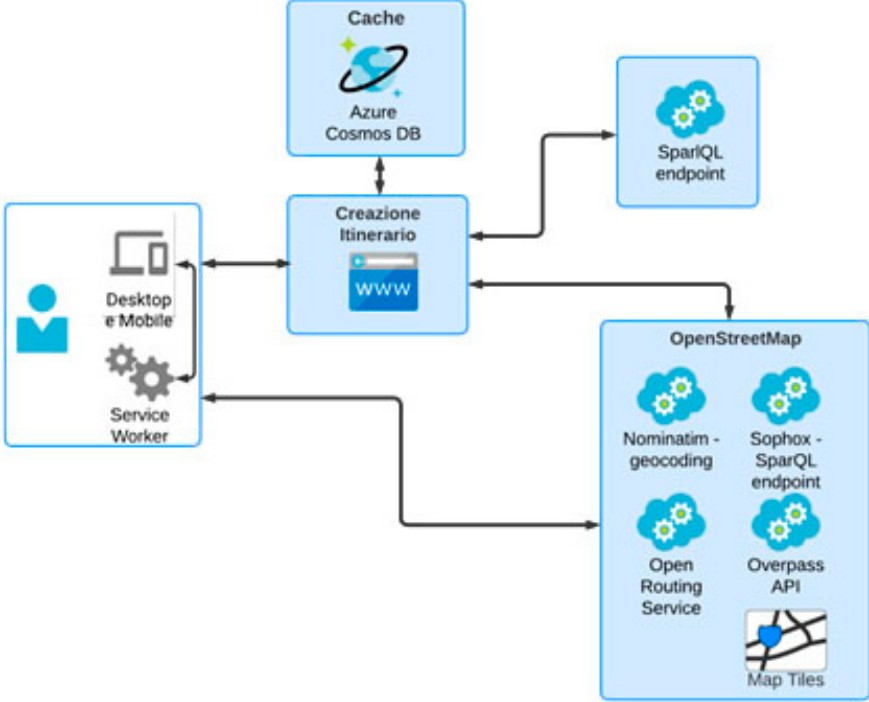

**Figure 3.** Euplea architecture.

This application is easily scalable horizontally, exploiting the Platform as a Service (PaaS) features of Herokuthat allow the number of instances of the application running to increase as traffic increases, while Azure Cosmos DB (4.2 server version) being a Software as a Service (SaaS) can scale in a completely transparent manner to the user, who only has to indicate the maximum level of service required.

Taking a closer look at the artefact, shown in Figure 4, two macro-areas can be identified: Single Page Application running on the client, i.e., on the device that the user uses to interact with the application, and REST API running on the server.

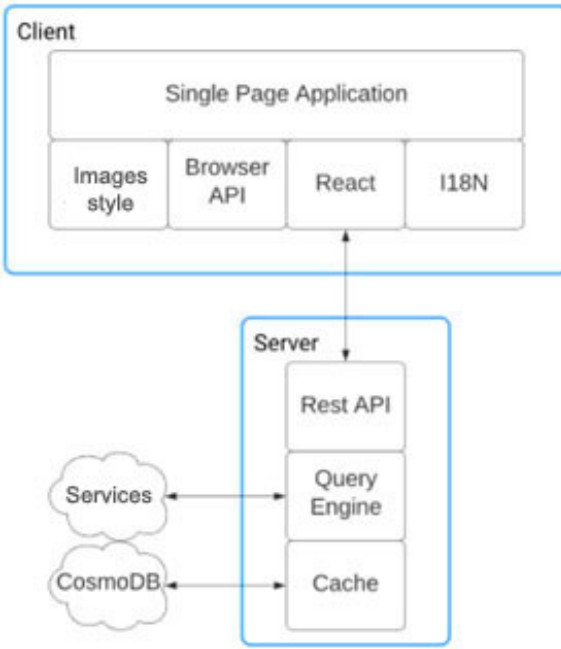

**Figure 4.** Artefact architecture.

The application takes care of returning to the user the Service Worker and the static resources such as JavaScript files and style sheets that make up the Single Page Application that will run on the client, as well as providing the REST API for requesting routes, and requires some environment variables to configure some of its functions.

The decision to use environment variables makes it possible to securely manage access credentials to the Cosmos DB database and allows for easy distribution. Furthermore, again with a view to verifying security and data integrity, communication between the client application running in the user's browser and the server is performed exclusively using the secure HTTPS protocol.

*3.1. Innovation*

A study was carried out to choose the web services and related communication protocols required to support keyword-based knowledge with existing search engine interfaces.

The use of the Linked Open Data (LOD) platform required that certain characteristics be considered in the choice:

- Use Uniform Resource Identifier (URI) to identify a physical or abstract resource;
- Use HTTP as the resource access protocol;
- Representation of data in Resource Description Framework (RDF) format;
- SPARQL protocol for querying;
- Inclusion of links to other URIs so that other resources can be discovered.

The SPARQL communication protocol for querying knowledge bases is thus defined by the choice of using LODs. SPARQL is an open-source standard, which, via the Internet

and a query and manipulation language, operates on data represented in RDF format in a similar way to how SQL is used for relational databases. The RDF data format is a standard for representing information in the semantic web and creating typed links between arbitrary objects (associations between resources) and consists of the triplet Subject–Predicate–Object [47].

Subject represents the main resource or entity about which a statement is being made, Predicate specifies the relationship or attribute between the subject and object, Object represents the value or object of the statement [48].

RDF data can be represented in several formats, including RDF/XML, JSON-LD, Turtle, N-Triples, N-QuadsTriG and TriX.

RDF triples, by exploiting semantic technologies and including metadata, can be combined to form more complex knowledge graphs, known as knowledge graphs. These improve the performance of search engines by providing direct answers to common queries, being able to combine data from different sources and extracting unsolicited information in keywords. In knowledge graphs, each node and each relation is represented by an RDF triple, and like RDF data, knowledge graphs can also be queried using the SPARQL language, searching for relations, patterns, related concepts, specific properties and more.

Knowledge graphs use the RDF model to represent a practical implementation of information and relationships between concepts in the real world that can incorporate not only structured data, but also semantic information and associative semantics between concepts.

By equipping the platform with a simple API, it becomes capable of connecting to various RDF databases, as well as to remote SPARQL endpoints for retrieving the information necessary for both the representation of cultural–historical itineraries and the storage of all the details and metadata required for data enrichment.

In addition, by exposing SPARQL endpoints in turn, the platform is able to offer the general public access to the data stored in it in various formats, as well as manually enriched or reconstructed data such as customized maps and archaeological and historical data annotated with metadata and georeferenced.

Tools capable of handling the types of data managed by the platform and retrieved from search engines and other external sources, such as structured and unstructured data, images, texts, etc., were also studied. MultiMedia Database Management Systems (MMDBMSs) capable of managing heterogeneous and unstructured multimedia data more efficiently than a traditional DBMSs and providing support to applications interacting with such information were then examined.

In particular, MMDBMSs must possess certain distinctive features:

- Support for multimedia data types, i.e., possess the ability to handle multimedia data such as images, audio, video and other multimedia formats.
- Functionality for creating, storing, accessing, querying and controlling multimedia data.
- Support for traditional DBMS functions, i.e., not only managing multimedia data, but also providing the traditional functions of a database management system, such as database definition, data retrieval, data access, integrity management, version control and concurrency support.

Among commercial MMDBMSs, a system of interest for the platform is Virtuoso Universal Server, of which there is an open-source version known as OpenLink Virtuoso. In addition to offering multi-model functionalities in a single system, allowing the management of structured, semi-structured and unstructured data, such as relational data, RDF, XML and plain text, the Virtuoso Universal Server also supports the SPARQL query language (in addition to SQL) and integrates file servers, web servers, web services and web content management systems. It also has built-in support for Dublin Core, a standard for describing and annotating digital resources by means of metadata that allows Dublin Core (DC) metadata to be associated with stored data.

DC metadata is a standardized set of metadata elements used to describe digital resources such as documents, images, videos, web pages and more. This metadata provides

essential information to enhance the discovery of digital resources, enabling users to find, identify and retrieve relevant content based on the information provided by the metadata. Some of the common DC elements are Title (resource name), Author, Date, Subject, Description, Format, Type, Identifier, Language, Publisher. These elements can be extended to include more detailed and specific information as required.

*3.2. Methodology*

At the beginning, the algorithms, and strategies necessary to correctly implement the application logic suitable for achieving the expected result were identified. Non-functional requirements such as security, performance, scalability, internationalization, maintainability and extensibility of the system were also considered.

The algorithms used to execute SPARQL queries depend on the management system that involve retrieving triples that correspond to certain patterns within the RDF graph; the pattern matching algorithms look for matches between the patterns specified in the query and the data in the graph. A key aspect in the execution of SPARQL queries is the resolution of triple patterns; triple patterns are parts of a SPARQL query that specify conditions on the subjects, predicators, and objects of RDF triples. The resolution of the algorithms is based on searching the RDF graph for triples that satisfy the criteria specified in the triple patterns. Optimization algorithms attempt to reorganize the query operations so as to reduce the number of triples to be examined or minimize the overall computational cost. Indexing structures to speed up access to triples during query execution exploit algorithms to create indexes on certain attributes (subject, predicate, object) to reduce search times. Some of the algorithms used in this work are:

- Join algorithms, when a SPARQL query involves multiple triple patterns, these algorithms are involved to combine the triples that satisfy each pattern.
- Caching algorithms, when some triple systems implement strategies to temporarily store the results of frequent queries and reduce the need to re-run the query.
- Parallelism algorithms, when the execution of SPARQL queries can be parallelized to take advantage of multi-core or distributed architectures.

It is important to note that triple store-specific implementations may vary and may use customized algorithms to optimize the performance of SPARQL queries. Mathematical solutions for SPARQL algorithms mainly involve concepts related to graph theory and relational algebra.

In graph theory, one can apply search algorithms such as Depth-First Search (DFS) or Breadth-First Search (BFS) to traverse the RDF graph.

Breadth-first search has a running time of $O(V + E)$, since every vertex and every edge will be checked once. Depending on the input to the graph, $O(E)O(E)$ could be between $O(1)$ and $O(V^2)$.

Relational algebra is a fundamental part of relational database theory. SPARQL has similarities with relational algebra in that it operates on triples (similar to tuples) and allows operations such as projection, selection and join. Relational algebra operators, such as the Cartesian product and the join operation, are key concepts in SPARQL when combining triples based on specific conditions

$$\Pi \, a, b \, (R)$$

$$\sigma \, a > 2 \, (R)$$

$$\bowtie \text{ full join, } \bowtie \text{ left join, } \bowtie \text{ right join}$$

In set theory, SPARQL often uses set concepts, since RDF triples can be seen as sets of tuples (subject, predicate, object). Such operations as UNION, INTERSECTION and DIFFERENCE in SPARQL can be understood through set theory.

Given a class of sets X (i.e., a class whose elements are sets), we define $\bigcup X$ (union of X) as the class of those objects that belong to at least one element of X. Then, $x \in \bigcup X$ if and only if $\exists a \in X$, such that $x \in a$. Another notation for $\bigcup X$ is $\bigcup A \in X \, A$.

First-order logic is involved in the definition of search conditions in SPARQL queries. WHERE clauses in SPARQL queries may include conditions based on first-order logic.

If A is a Well-Formed Formula and X is a variable, $\forall X\ A$ and $\exists X\ A$ are Well-Formed Formulas.

Data structures such as trees and graphs, queries as shown in Figure 5, can be used to represent and organize RDF data, facilitating the implementation of efficient algorithms for SPARQL.

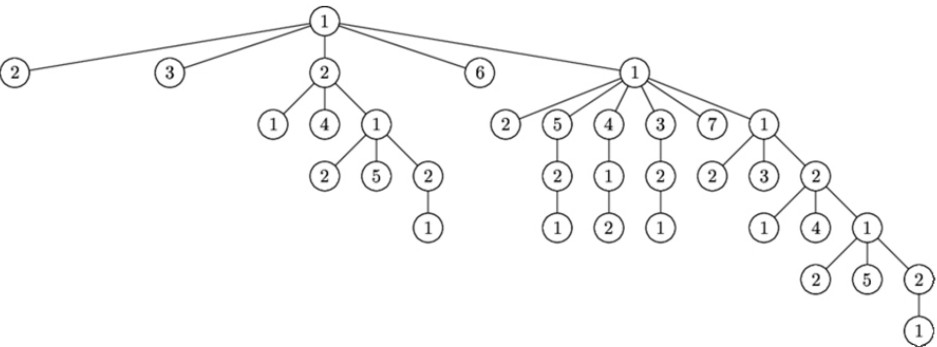

**Figure 5.** Example graph.

In addition, in the definition of the reference architecture and the implementation of the solution, the problems relating to the execution and distribution of an application which, by its nature, will have to use different services and datasets on the Web and will have to work both on desktop and on smartphones and tablets.

By integrating data on cultural heritage with geographical information, made available by the OpenStreemap project via GeoSPARQL queries, it will be possible to define an itinerary with appropriate indications for overnight accommodation and refreshments [48].

Having identified the points of interest for the itinerary, it will be possible to calculate the optimal itinerary using the user's position with the specified duration within the selected geographical area and show the result on the map as shown in Figure 6.

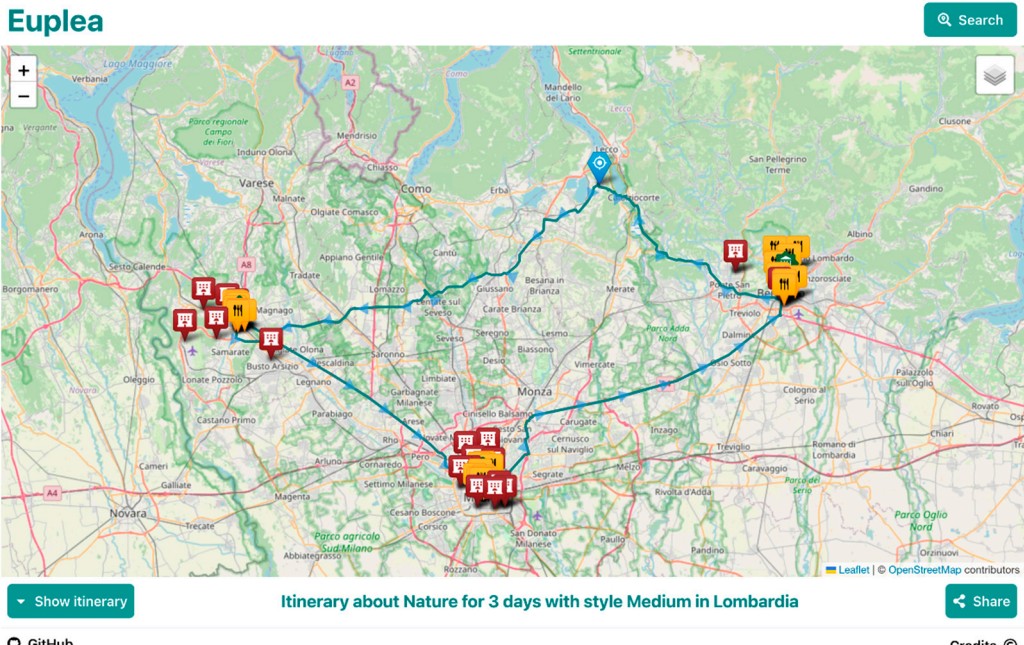

**Figure 6.** Map result after identification of points of interest.

It will also be possible to see the detail of the itinerary obtained, with an indication of the number of relevant cultural assets identified as shown in Figure 7, and share it with other people.

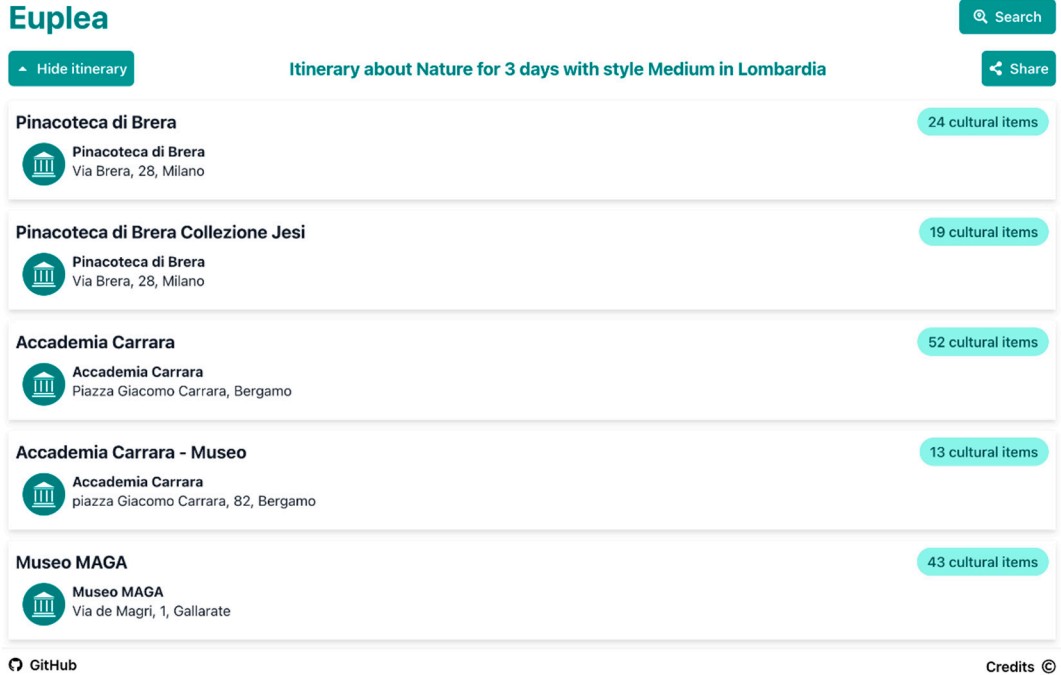

**Figure 7.** Map results.

### 3.3. Selection of Places of Interest

To select places of interest, a process consisting of several steps has been implemented, which, starting from a topic of interest, the possible indication of an Italian region and the number of days available, is able to offer the user a series of places to visit, as illustrated in Figure 8.

To start the procedure, it is possible to indicate a particular topic of interest to carry out the search, possibly indicate the Italian region to narrow the search and the number of days available to limit the number of locations proposed by the itinerary. Using a SPARQL query, all places are identified, which preserve cultural objects relating to the topic of interest with an indication of the number of objects identified; the results will be sorted in order to propose the places with the greatest number of objects, as detailed in the following sections, which use the Italian regions identified with a different SPARQL query.

Unfortunately, the structure of the data used turned out to be non-homogeneous; therefore, to identify the geographical coordinates of the identified places, it was necessary to analyze various relationships present in the database and use a geocoding service to obtain the correct coordinates.

This type of query was very slow for the public SPARQL endpoint used; response times are in the order of ten seconds, higher than the limit of 2 s that a Web user is willing to tolerate [49]. Therefore, it was necessary to implement a persistent cache system that allows faster access to the results.

Once the places have been obtained, with the indication of their position and the number of interesting objects preserved, it is then possible to group them by the cities in which they are preserved, in order to be able to identify the cities that preserve the greatest number of cultural assets; we continue with the identification of services and accommodation available nearby through geographical queries, as will be detailed below.

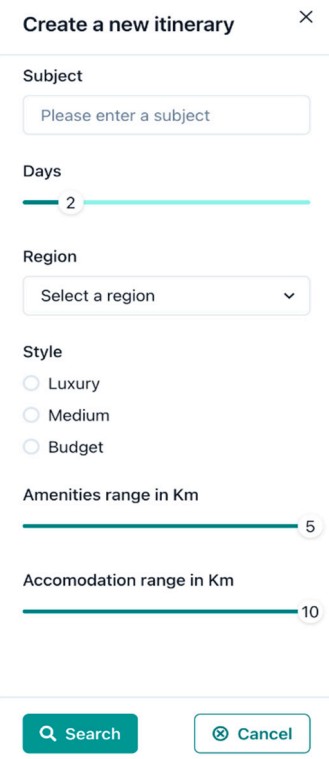

**Figure 8.** Selection of places of interest.

### 3.3.1. Search for Culture Points

By analyzing the data contained in the Catalog of Cultural Heritage, the CulturalProperty class of the Arco ontology was identified; this class represents both a material and immaterial cultural asset, recognized as part of the national cultural heritage and useful for the knowledge and reconstruction of history and landscape. For the query, all the subclasses that possess the searched topic were considered.

Once the cultural assets have been selected, considering the relationship hasCulturalInstituteOrSite [50], defined in the Location ontology of Arco [51], which connects a cultural asset to its container (place or institute of culture), we can obtain the places where they are preserved. Those that are deprecated by the presence of the owl:deprecated value are automatically excluded.

By possibly filtering by the selected region, grouping the result obtained by cultural place and counting the cultural objects, the requested data were extracted together with the indication of the city in which they are located.

### 3.3.2. Identification of the Italian Regions

In the previous section, the cultural assets were filtered by region to which they belong. For this reason, it was first necessary to identify which were the Italian regions. From an analysis of the data, the regions were found not to be uniquely represented in the Cultural Heritage dataset, but present the critical issues already highlighted.

### 3.3.3. Non-Homogeneous Data

A very obvious example concerns geolocation, in which two ontologies Location Ontology and Basic Geo (WGS84 lat/long) Vocabulary [52] are used to represent latitude, longitude and other spatial information. Geolocation information was associated inconsistently, using different data hierarchies.

These issues require the implementation of more complex queries that are capable of managing multiple different ontologies on the one hand, and examining different data structures to extract information on the other. Clearly, as the number of different strategies used in the query increases, the system proportionally requires significantly more time.

In the presence of non-homogeneous data, it is not possible to guarantee the exhaustiveness of the data obtained, because it is not possible to implement an exhaustive query, as neither the number of different ontologies used nor the hierarchy of classes used are known a priori.

For example, within the dataset, the geographical coordinates of a CulturalInstituteOrSite can be obtained in various ways:

- Directly from the lat and long coordinates of the Basic Geo ontology;
- Through the hasSite relation which takes us to the Site class which has the lat and long coordinates of the Location Ontology;
- Through the hasGeometry relation of the Italian Core Location Vocabulary (CLV) ontology [53] towards the Geometry class, which has lat and long coordinates of the Location Ontology;
- Through the hasGeometry relation of the CLV ontology towards the Geometry class and then through the hasCoordinates relation of CLV towards the Coordinates class, which has lat and long coordinates of the Location Ontology;
- Using the coordinates associated with a cultural property to which the same siteAddress class is associated, associated with the Site, associated with the CulturalInstituteOrSite class.

After querying these sources, we need to merge all these results into a single pair of lat and long coordinates so that we can return the information.

To avoid exceeding the maximum execution time of a SPARQL query in the database, it was necessary to execute the individual queries separately; subsequently, it is implemented within the application the logic to identify the geographical coordinates or, in their absence, the address to proceed with geocoding.

### 3.3.4. Geocoding

To obtain the spatial coordinates absent in the Cultural Heritage dataset, it is necessary to use Geocoding; available information are used (name of the site, address of the site), which have similar problems to those previously exposed.

In order to obtain good quality results, for each site subjected to geocoding, three separate geocoding operations are carried out. Different values and parameters are used to then sort the results obtained in accordance with the importance parameter returned by the service itself.

The geocoding service used is OpenStreetMap's Nominatim, which has a limit of one request per second. To limit excessive consumption of service resources, the results are stored in the application cache.

### 3.3.5. Itinerary Creation

Once a list of geolocalized cultural places has been obtained, ordered by number of cultural assets, the stages of the itinerary must be defined. To simplify the algorithm, it was assumed that only one city could be visited per day and a parameter was introduced to select how many cultural places can be visited each day. With these simplifications, a recursive algorithm has been implemented that implements the following logic:

- Group cultural places by city and sort the list in descending order;
- Remove the first N cultural places with multiple objects from the list to create a stop on the itinerary, where N is the maximum number of cultural places that can be visited per day;
- Repeat the algorithm if cultural places are still present.

From this list, it is then sufficient to select the number of days to obtain the requested itinerary.

### 3.3.6. Selection of Services and Accommodation

To create an itinerary that can be used by users, once the sites to visit have been identified, it is necessary to provide information relating to the services available and the accommodation available for overnight stays. Selection criteria were therefore implemented, such as style, which determines the type of services and accommodation to be shown, and the distance from places of interest, as in Figure 9.

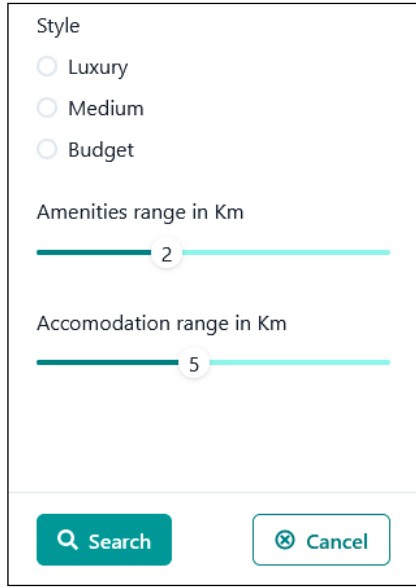

**Figure 9.** Selection of services.

Using OpenStreetMap data, the search for services and accommodation is carried out with two alternative implementations, one based on Sophox and one on Overpass.

For both services, it was possible to use the same criteria as they both rely on the same database.

For style-based accommodation selection, hotels are divided according to their stars.

Other types of accommodation were also considered: 'chalet', 'apartment', 'hostel', 'guest_house', 'motel', 'camp_site', 'alpine_hut', 'wilderness_hut'.

As regards the selection of services, the classification shown in the following code fragment was used:

Style.Luxury homes:
return ['restaurant']
Case Style.Medium:
return ['restaurant', 'food_court', 'pub', 'cafe']
Style.Budget homes:
return ['pub', 'cafe']
Sophox is queried via GeoSPARQL queries; in addition to the implemented filters, using the illustrated information, the distance in kilometers from a point is indicated.

Overpass querying was implemented using Overpass QL; in addition to the filters implemented, using the information illustrated, the distance in meters from a point is indicated.

### 3.3.7. Cache

For all the services mentioned, a simple persistent cache system has been implemented both to improve their use and to respect the conditions of use of the services themselves.

Before sending the request to the service, the cache is queried and if the response is already present in the cache, the value present in the cache is returned without making the request to the service. If the value is not present in the cache, the request is instead sent to

the service and its response, and even in the event of an error, is saved in the cache before being returned.

To avoid showing users stale data, a Time To Live (TTL) control has been implemented, which clears the cache when it is old.

To maximize performance for each type of request, a separate container is used, created on the first request, indexed using the same field as the unique identifier of the request.

The cache data is saved in the Microsoft CosmosDB system [54], which offers:

- NoSQL Database features with a simple API;
- An availability level of 99.999%;
- A free level of service more than adequate for the needs of this project.

All the various services have been implemented in such a way as to make it possible to use the application without a cache if the credentials for using CosmosDB are not provided.

### 3.3.8. WebGIS Presentation

The generated itinerary is shown in Figure 10, using Leaflet together with Open-StreetMap cartography.

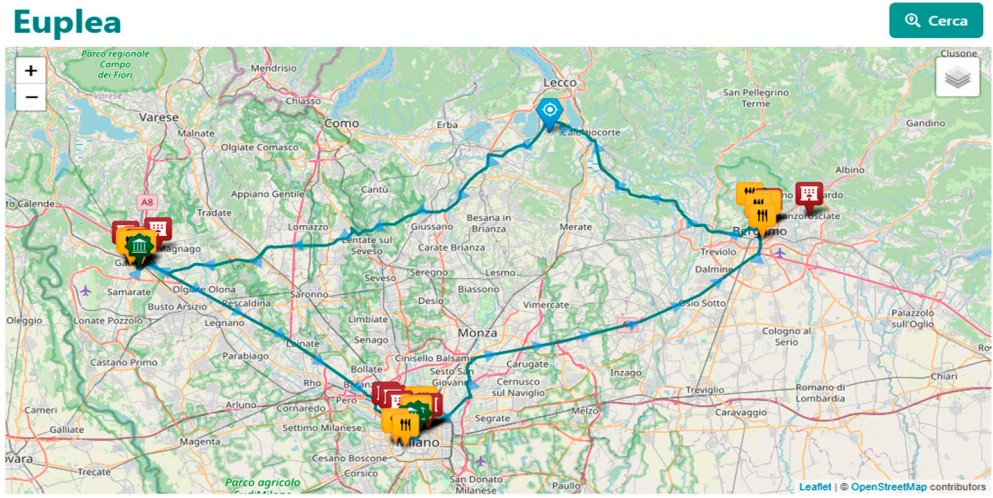

**Figure 10.** The map presents the required services.

### 3.3.9. Layers

There are different levels in the map, updated independently to increase the performance and responsiveness of the application; information is displayed progressively as it becomes available.

The different levels are as follows:

- Current position: the current position, represented with a blue icon, is obtained using the Geolocation API [55], supported by all the main browsers, asking the user for permission;
- Raster map: the OpenStreetMap service provides the tiles used for the map;
- Cultural Sites: the identified cultural places are represented with a green icon;
- Accommodations and services: the accommodations and services are extracted as indicated and are represented in two separate layers; they are indicated, respectively, with a red and a yellow icon, and can be deactivated using the control at the top right;
- Itinerary: the itinerary to visit all the cultural places shown constitutes a polygon separated from the other levels, and is decorated with triangles to indicate the direction of the route.

To avoid duplication in the code, the layers relating to the Cultural Sites, Accommodations and Services were implemented using the same generic software capable of representing a set of points of interest.

### 3.3.10. Routing

The polygon, which represents the optimal sequence with which to visit places of interest, is generated using the OpenSource Routing Machine service, providing the coordinates of the Cultural Sites and, if available, the user's location [56].

## 4. Conclusions and Future Work

### 4.1. Conclusions

As previously illustrated, the development of Web technologies has provided us with powerful tools, such as Linked Open Data, to add a semantic layer to data which, thanks to WebGIS technologies, allow us to easily create new applications by introducing new innovative means of using data.

In addition, the development of features such as Service Worker and Progressive Web App allow us to extend the Web beyond its traditional boundaries, making a Web application indistinguishable from a native one and eliminating the complexity of having to manage each native platform differently.

The availability of cloud services "as a Service", widely used in this project, both to develop the application and to host it, as well as to retrieve the data necessary for its operation, lowers the cost of realizing solutions of this type and further enriches the ecosystem of the Web by making it increasingly ubiquitous and pervasive.

It should also be considered that the creation of a Web application using various other Web services is greatly conditioned by the performance of the services themselves and the type of network connection linking them. Although various strategies have been employed to increase performance, such as the introduction of a persistent distributed cache and the use of Service Worker and Single Page Application, it is not possible to control the performance and availability of third-party systems from which data is requested.

With WebGIS technologies, it is possible to easily create new applications which, by adding a semantic level to the data, allow us to introduce new tools for use.

A Web application is indistinguishable from a native one with the advantage of eliminating the complexity of having to manage each native platform differently.

The availability of cloud services "as a Service", widely used in this project, to develop the application, to host it, and to find the data necessary for its operation, lowers the cost of implementation and further enriches the ecosystem of the Web, making it increasingly omnipresent and pervasive.

Linked Open Data presents critical issues as already specified.

It should also be remembered that a Web application uses various other Web services, and is significantly influenced by the performance of the services themselves and the type of network connection.

Although various strategies have been undertaken to increase performance, such as the use of Service Workers and Single Page Applications, it is not possible to control the speed of response.

### 4.2. Future Developments

This project showed how Linked Open Data can be successfully used and represented via WebGIS. Some further possible developments aimed at expanding and improving the application are illustrated below:

- Use additional Linked Open Data sources.

For this project, the Linked Open Data made available by the Ministry of Cultural Heritage and OpenStreetMap were used, but it may be interesting to consider additional datasets to enrich the itineraries.

One direction could be to evaluate the cultural assets of other countries, to extend the available territory beyond national borders.

At the same time, it could be interesting to use other sources of information; for example, by drawing on the Linked Open Data made available by the Wikipedia project.

Other types of Linked Open Data could be added to show on the map and provide other types of information to the user; for example, data relating to the environment or weather conditions.

- Introduction of other selection criteria.

It would certainly be interesting to increase the range of filters available when defining the itinerary. It could be interesting to allow the user to filter by historical period, by artistic current, by type of cultural asset, by other interesting criteria present in the ontologies used.

- Display itineraries created as Linked Open Data.

A further development of this project could deal with exposing the itineraries created as Linked Open Data. The ontologies created within the Cammini D'Italia project could be suitable for describing this type of data, thus avoiding having to define a new ontology.

- Analysis of the itineraries created and user feedback.

We cannot fail to mention the development area relating to the analysis of the data of the generated itineraries. Information could be obtained on the most sought-after cultural assets, as well as on the most requested regions, on the duration and on any other filters made available.

It might also be interesting to create a feedback system for users, so that they can report anomalies and incorrect data. By analyzing the reports, it would be possible to improve the algorithm for creating itineraries. Directly, with the permission of the data owner or by submitting a report. the data present in the Linked Open Data can be updated.

- Integration with virtual assistants.

Thanks to the choice to create REST APIs, which contain the logic for creating the itineraries, it would be possible to reuse them by creating an application that extends the functionality of virtual assistants; for example, by creating a Skill for Amazon Alexa. In this scenario, the user could request an itinerary by vocally querying the virtual assistant.

Alexa could provide the appropriate selection criteria, and the assistant would respond by showing and describing the places that make up the itinerary.

**Author Contributions:** Conceptualization, all authors; methodology, Mauro Mazzei; software, Matteo Spreafico and Mauro Mazzei; validation, Mauro Mazzei; formal analysis, Mauro Mazzei and Matteo Spreafico; data curation, all authors; writing—review and editing, Mauro Mazzei and Matteo Spreafico; supervision, Mauro Mazzei All authors have read and agreed to the published version of the manuscript.

**Funding:** This research received no external funding.

**Data Availability Statement:** OpenStreetMap data is licensed under the Open Data Commons Open Database License (ODbL). This area is too large to be exported as OpenStreetMap XML Data.

**Acknowledgments:** The authors thank the projects of the CNR institutes as a source of documentation necessary for the conduct of this paper.

**Conflicts of Interest:** The authors declare no conflicts of interest.

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
