# Peer review of "Cultural Itineraries Generated by Smart Data on the Web"

_ijgi, doi:10.3390/ijgi13020047_

Round 1
Reviewer 1 Report
Comments and Suggestions for Authors
I have studied the proposed manuscript in detail. My opinion on it boils down to this: I think it should be edited as far as the introduction is concerned, as well as the text should be refined in terms of structure.
For example, in the Introduction there are enumerations, tables, and general parts that could actually be summarized in a separate part that, together with the literature review, would focus on the essence of web technologies for data sharing. I will also point out something specific - at the very beginning it is not necessary to list the principles here is not necessary; they can also be presented more tightly with a focus on the specifics under consideration. There are other such examples. In my opinion, they should be cleaned up and the Introduction should once again present the focus of the paper, the context the authors consider, the hypotheses they talk about in the Abstract, and the results they expect.
I will allow myself to make a similar comment regarding the methodology chosen for the implementation of the study. It is definitely a case study analysis. This should be emphasized by justifying why it is actually the right choice and only then, to present the details as it is done in the subsequent parts of its description - in 3.2 Methodology.
Reading the text, I get the impression that the goal of verifying “the potential of the use of Linked Open Data in the world of WebGIS and illustrate an application that allows the user to interact with Linked Open Data through their representation on a map” has been achieved. However, I would recommend that the conclusions (in this regard) be presented more clearly. Also, I consider it appropriate the style of writing in the Conclusion to be corrected by avoiding expressions like: (in the sentences) “In 2007, Jeff Atwood enunciated what we now know as Atwood's Law: "Any appli-cation that can be written in JavaScript, will eventually be written in JavaScript." “Today, fifteen years later, we can certainly see that this law based on Tim Berners-Lee's The Rule of Least Power is increasingly being respected.” or “The development of features, such as Service Worker and Progressive Web App, al-lows you to extend the Web beyond its traditional boundaries.” (all in the Conclusion part).
Author Response
Thanks to the suggestions, we have included in the first part an introduction focusing on the approaches we used and the objectives we achieved. We have also motivated the choices made. We have revised the entire first part by adding the choice of algorithms used at the beginning of the proposed method. As for the conclusions, we decided to remove the part of the references and citations.
Reviewer 2 Report
Comments and Suggestions for Authors
The topic of the work is very interesting and current and excellently handled from the professional point of view.
However, the scientific component is completely omitted from the work. The bibliography is also very modest as far as scientific works are concerned. The authors should add an overview of the existing scientific literature related to the research topic and emphasize the scientific contribution of the work, since it is a publication in a scientific journal.
Author Response
Thank you for your comments. We have revised the entire first part by adding the choice of algorithms used at the beginning of the proposed method, which is the most relevant scientific contribution of the proposed work.
Reviewer 3 Report
Comments and Suggestions for Authors
The title is interesting.
The work concept is good.
Literature is appropriate and innovative.
Methodology is well described.
Images adequately represented.
The results and discussion are described.
The paper contributes to the development of science and I recommend it to the editors for publication.
Good luck in your future work.
Author Response
Thank you for your comments, some initial parts have been expanded upon, see introduction and methodology.
Reviewer 4 Report
Comments and Suggestions for Authors
I recommend a better description of the Geodata presentation in the study and connection methods. As Figure 3, the design tools use only from open street map database.
The article will be published in an English journal, then it's better to use english words in all figures.
the designed site will provide a service to see the details of the itinerary obtained and it's necessary to add English mode for non-Ialian users.
Author Response
Thank you for your comments, some initial parts have been deepened, see introduction and methodology. We have changed the figures to correct language.
Round 2
Reviewer 2 Report
Comments and Suggestions for Authors
I have no further comments, the paper is much better now, I recommend publication.